# Modeling of Vanished Historic Mining Landscape Features as a Part of Digital Cultural Heritage and Possibilities of Its Use in Mining Tourism (Case Study: Gelnica Town, Slovakia)

**Pavel Hrončok [1], Bohuslava Gregorová [2], Dana Tometzová [1],\*, Mário Molokáč [1] and Ladislav Hvizdák [1]**

[1] Department of Geo and Mining Tourism, Faculty of Mining, Ecology, Process Control and Geotechnology, Institute of Earth Resources, Technical University of Košice, Letná 9, 042 00 Košice, Slovakia; pavel.hroncek@tuke.sk (P.H.); mario.molokac@tuke.sk (M.M.); ladislav.hvizdak@tuke.sk (L.H.)

[2] Department of Geography and Geology, Faculty of Natural Sciences, Matej Bel University in Banská Bystrica, Tajovského 40, 974 01 Banská Bystrica, Slovakia; bohuslava.gregorova@umb.sk

\* Correspondence: dana.tometzova@tuke.sk

**Abstract:** The study provides a methodology for 3D model processing of historic mining landscape, and its features as mining digital cultural heritage with the possibility of using new visualization means in mining tourism. Historic mining landscapes around the towns of Gelnica (eastern Slovakia) had been chosen for the case study. The underground mining spaces around Gelnica, which are currently inaccessible to clients of mining tourism, were processed using 3D modeling. Historically, correctly processed 3D models of mining spaces enable customers of mining tourism to virtually travel not only in space, but what is most important, in time as well. The up-to-date computer-generated virtual mining heritage in the form of 3D models can be viewed via the Internet from different perspectives and angles. The models created this way are currently the latest trend in developing mining tourism.

**Keywords:** 3D modeling; digital cultural heritage; mining tourism; vanished and inaccessible mining features

## 1. Introduction

During the Middle Ages and the Early Modern period, the territory of Slovakia within Hungary belonged to the most important mining sites of the world. A number of these sites have been preserved in good conditions, making it ideal for the development of mining tourism. The town of Banská Štiavnica, which has a similar character, had been added to the UNESCO World Heritage List (http://whc.unesco.org/en/list/) due to its mining monuments in 1993. There are several other interesting historical mining sites in Slovakia like Gelnica town, where mining tourism develops only very slowly.

3D visualization with the use of modern tools is very important, and in many cases, irreplaceable in tourism development, as well as its presentation and promotion. It can increase the attractiveness, potential, and visitor attendance of mining tourist sites (also in mining tourism). Increasing the attractiveness is justified in case of preserved, physically existing landscape environment of the sites. In the case of non-existing or inaccessible mining locations, 3D visualization can boost the attractiveness or can create a new potential of these sites. The inaccessibility could be caused by unsuitable technical conditions. Then underground areas are secured against unauthorized entry,

eventually, they are purposely buried or blasted, etc. There is also frequent legislative barriers in the form of legal regulations and restrictions (Mining Act, Acts on the Protection of Monuments and Landscape, etc.) or ownership relations. Today, 3D modeling is booming worldwide in all areas of society, including (mining) tourism. It is widely used in the tourism industry, especially while displaying non-existing (demolished) cultural monuments in situ [1–6].

The whole study was created for the needs of mining tourism as a new and increasingly developing form of tourism. Mining tourism is starting to fill one of the gaps in contemporary modern tourism. In recent years, the opportunity of old, abandoned, but still operated mines and active mining regions has become an important additional touristic element in many regions worldwide.

There are more and more mines in various parts of the world that are turning into tourist attractions. There are several reasons for doing so, such as the possibility to revive and manage closed, inactive mining facilities. The reason is also the possibility to use the potential of people living in former mining towns and their knowledge of the mining industry. Finally, in the era of constant dynamic development of tourism, we can see the trend of tourists, who are constantly longing for new adventures, searching for new attractions. Today's tourists, as well as ever globalizing experts, travel a lot, willingly visiting new destinations and, most importantly, unusual and sophisticated attractions. It should be emphasized that, especially in recent years, significant development of new products and tourist attractions have been observed in previously unrecognized mines or mining regions of the world. These objects are located underground and can be perceived as exceptional, distinctive, and unique peculiarities specific for a particular destination [7].

An interesting way of increasing the attractiveness of mining sites for the clients of mining tourism is 3D visualizations of the individual mining objects and their subsequent availability on the Internet. A lot of historical monuments are currently digitized and presented online. The paper, however, points to the possibility of creating 3D models of the already non-existent or inaccessible historical mining structures and increasing the potential of the mining sites. This approach requires not only high-quality programmers and graphic designers but also skilled scientists able to create historically relevant descriptive texts, diagrams or models using which these objects can be computerized for the use in mining tourism. The 3D visualizations that are based on archival research are irreplaceable in those cases where mining objects and elements no longer exist in situ, and, by this method, it is possible to create their real computer reconstructions as part of digital mining cultural heritage.

The aim of the paper is to process the positive and negative aspects of 3D visualization use in mining tourism. The article also points out the benefits of 3D visualization through mining models of the non-existing (or inaccessible) historical mining underground spaces on the example of Gelnica town.

## 2. Study Area

Historical case studies of the mining landscapes of Gelnica's surroundings have been used for the needs of this work. The selected historic mining town, which previously belonged to the Union of Upper Hungarian Mining Towns, is located in the south-eastern part of Slovakia in the Košice Region (Figure 1).

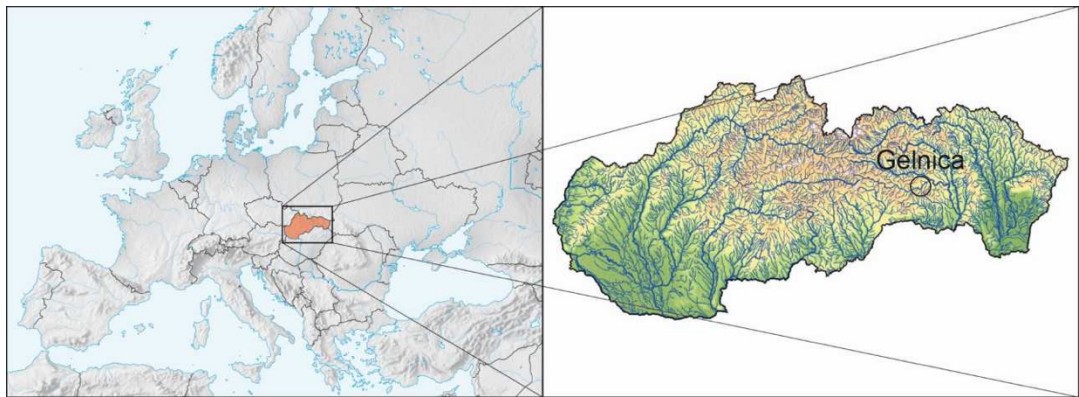

**Figure 1.** Location of the mining town Gelnica, within Slovakia and Europe.

The Union of seven mining towns of the Upper-Hungary was officially created on the 26 December 1478 in Košice and had been formed by the towns Gelnica, Jasov, Rožňava, Smolník, Spišská Nová Ves (nowadays located in Slovakia), Rudabánya and Telkibánya (Hungary). All of these towns had adopted the Gelnica Law by the oldest and the most important town of this union—Gelnica.

The most important ore veins of the Gelnica district are the veins Gelnická, Krížová, and Boží dar. All of them had been excavated in the late 13th century, initially because of the silver, later because of copper as well. Around the half of the 19th century began a gradual decline of mining, which ended in mine closures in the early 20th century. The individual veins are interconnected and stretch in the hinterland of the valley of the river Hnilec, in a total length of over 20 km [8,9].

## 3. Materials and Methods

Mining heritage is a specific complex of tangible and intangible monuments which are directly or indirectly related to the raw material extraction (mines, technical facilities, buildings, etc.), miners (their daily routine and life, society, clothing, health, etc.) and also intangible elements (religion, customs, songs, traditions, etc.). These elements form a separate system of tangible or intangible values important for today's society and deserve to be preserved for future generations [10–14].

The territory of today's Slovakia (the Western Carpathians) was one of the most important mining areas in the world in the Middle Ages and early modern times. Many of these landmarks have been lost forever, but their restoration is possible in accordance with the UNESCO Charter on Preservation of Digital Heritage. The study presents the given methodology as one of the possibilities of mining cultural heritage restoration. The disappearance and loss of such landmarks would be otherwise seen by the Charter as the impoverishment of the heritage of all nations.

The following principles apply to digital mining cultural heritage restoration and preservation in accordance with the UNESCO Charter on Preservation of Digital Heritage [15]:

- Mining digital heritage consists of unique sources of human knowledge created digitally or converted to a digital form from existing analog sources. If the source is born as 'digital', there is no other format but the digital object given.
- The most valuable materials of digital mining cultural heritage are 3D models and visualizations.
- The purpose of preserving digital mining heritage is to keep it accessible to the public. Mining tourism seems to be a suitable mass form of heritage accessibility to the general public.
- Digital (including mining) heritage should be as one of the key elements of national heritage conservation policy, included in archival legislation and compulsory deposits of libraries, archives, museums, and other public repositories.
- Considering time, geography, culture and format, digital (including mining) heritage is naturally unlimited. It is potentially accessible to everyone in the world.

- There should be a state-appointed agency responsible for coordinating the preservation of digital heritage.

Two aspects are a priority for further tourism development. The first one is the implementation of new information technologies at all levels, while virtual mining tourism fully develops this aspect in terms of the creation of mining cultural heritage virtual products and their digital use by the participants. The second aspect is related to the unjustified enlarging of recreation areas in the open landscape. While in this case, the priority is the use, restoration, and revitalization of existing areas with objects in the form of cultural (mining) monuments, with the aim of their protection, preservation, and accessibility to the general public [10,14,16–18].

The interest in mining tourism is associated with the motivation of tourists and their focus on a specific goal. Tourists' motivation is connected with the environment devoted to mining tourism. If a mine or a mining site is in full operation or it can pose security risks, the hazardous environment can lead tourists to risky adventures. An important step for the development of mining tourism is, therefore, the adaptation of the mine to the needs of tourism. Costa and Santos (2016) describe a natural, logical sequence in three steps of turning an active mining plant into a tourism destination: Mine in operation—Discontinuation of mining activities and mine closure—Adapting mine to mining tourism [7,10,16,19]. Moreover, they see the future of mining tourism in the creation of thematic montane parks based on the model of geothermal parks, geoparks, or various cultural parks.

When it comes to historical mining sites with almost no tourist attractions that are no longer in use and which underground space is flooded or inaccessible, there is a large space for the development of computer modeling and visualization (surface and subsurface), technical equipment and processes. In this modern trend in mining tourism, digital mining heritage, its creation, preservation, and presentation play an important role.

Mining tourism is a form of adventure and cognitive tourism for specialists and the general public. The interested person in mining tourism can take advantage of a combination of both experiences and knowledge of visiting in-situ mining sites and regions, visits of mining museums, and from literature and archive studies, including mining documentation. In situ mine visits helps a tourist get to know used mining technologies and processing methods of raw materials throughout history. Visits of mining regions help tourists to understand the boom and bust cycle of the mining region, and to learn the habits of the miner community in different times of history.

The first definitions of mining tourism come from Pavol Rybár and his colleagues [13,20,21] —'Mining tourism is a form of adventure tourism, where the presence of a tourist in underground mining areas is providing him with new feelings and sensations. Mining tourism is defined as a phenomenon describing unique mining machinery and facilities, enabling exploration of the underground spaces with specific abiotic and biotic components, allowing one to admire the cultural heritage linked to historical mining, which is opened to the general and professional public'.

A much more general view of mining tourism is offered by Kršák and his colleagues (2015): 'Mining tourism brings together the aspects of industrial, technological, cultural and ethnographic heritage into a cognitive-educational-experience oriented form of tourism' [22]. Products of mining tourism are considered from the perspective of tourism destinations as components of the destinations' offer. A similar view is offered by Różycki and Dryglas (2017): 'Mining tourism is any form of tourist activity in industrial sites, technological sites, and industrial heritage sites. Most frequently, these sites are carefully prepared as tourism products [7].

Authors Costa and Santos (2016) understand mining tourism as the product of geological heritage (which is a working object of geotourism) and industrial heritage (which is a working object of industrial tourism) [19]. A basic explanation of the interest of mining tourism (some authors use the term "montane tourism") has been mentioned in many publications [14,16,17,23–25].

Virtual technologies have changed the way people perceive the landscape and the world around them. Various studies show that the population has generally become very quickly familiar with the technology of the modern and digital realities, whether virtual or augmented [26].

Thanks to modern technology, data can be generated by various multimedia devices (computers, tablets, smartphones, etc.) and overlaid with elements of the real world, thus making the user's perception even stronger [27].

The integration of digital reality technology into tourism enables operators to bring new and more interesting packages for tourists ever. Virtual reality leads to the creation of multimedia packages according to the user's preferences based on various scenarios that a tourist attraction can provide. The presentation of (montane) elements and processes in the digital environment must be professionally correct and as accurate as possible so that the end result is natural as a whole for a given historical period. Digital content must upgrade tourists' experiences [28].

The technology and system of augmented and virtual reality enrich the real world with virtual objects and scenes such as images (2D/3D), videos, sounds, etc., in the real world. It comprises a variety of technologies, including image processing, computer vision, computer graphics, and a display. It evokes a sense of presence in the real world, or more precisely, in a synthetic environment that is characteristic for virtual reality and allows the user to get to know the surrounding space of the montane landscape that is no longer used for mining in different time periods [29]. It also allows tourists to enter an inaccessible mining underground, or to see underground elements that cannot be detected with the naked eye (e.g., ore veins and their reflection in the mining underground as stated in the study).

As a result, virtual reality is a unique tool for tourism and cultural heritage, as various applications provide an opportunity to view and tell the stories of the places and history of the montane sites under discussion.

Virtual reality offered users an interactive simulated montane environment. Its main disadvantage in mining tourism is that it does not allow a tourist to make a connection with the real environment in its surroundings as it requires full immersion in the simulated environment. Unlike simulated reality, shared reality enables this connection because one of its prerequisites is to apply computer-generated data to a real-world view. This is perhaps one of the main factors in increasing the popularity of shared reality in mining tourism [28].

Virtual Reality (VR) offers tourism to many useful applications that deserve more attention from researchers and tourism professionals. With the continued development of VR technology, the number and importance of such applications will undoubtedly increase. Planning and management, marketing, entertainment, education, accessibility and heritage protection are six areas of tourism in which VR can prove to be particularly valuable. The possible use of VR as a source conservation tool is based on its potential to create a virtual experience that tourists can accept as a substitute for real visits to endangered sites. However, the acceptance of such alternatives will be determined by tourists' attitude to authenticity, and by their motivation and limits. As VR continues to integrate into the tourism sector, new issues and challenges will emerge. The industry will benefit from future research on the topics under discussion and will present a number of proposals for future research [2].

The geological maps and schemes and the old mining maps had to be georeferenced (correctly placed in the current system of coordinates) and then combined into a single image in order to obtain complete information throughout the sites [30–32]. ArcMap and ArcCatalog, which are parts of the ArcGIS 9.1 software package, had been used for maps georeferencing within a geographic information system [33]. GPS coordinates of the four selected points were transformed into the coordinate system using the JTSK ConvertCoord application, normally used by surveyors to transfer GPS data (latitude and longitude) coordinates to JTSK coordinate system, which is required for software data processing using GIS to ensure the base map is placed in the correct position.

By the creation of 3D models for the needs of geotourism and mining tourism, we followed the methodology developed by Molokáč and Hvizdák, in collaboration with prof. Rybár at the Faculty BERG of the Technical University in Košice [34–40]. According to the methodology, the creation of 3D model as a visualization tool of mining landscape and mining underground for the purposes of mining tourism can be divided into several main areas: digital data provision (ensuring conditions necessary

to obtain quality digital data using a digitizing device), collection of identical points necessary for georeferencing, georeferencing, the possibility of displaying a digital map and underground objects in a 3D model, visualization (description of the digital georeferenced data processing leading to 3D model creation) and accessibility (the presentation of online accessibility options for the 3D map and underground model).

The 3D data collection is of particular interest to geodesy, photogrammetry, and Earth remote sensing. Just a few years ago, the aerial and ground imagery, large scale maps, and direct geodetic measurements were common data sources.

LiDAR (Light Detection and Ranging) technology based on 3D laser scanning (aerial and ground) as well as high (sub-meter) resolution satellite imagery is now increasingly being used. One of the key elements of 3D modeling is the geometric (morphological) aspect represented by the Digital Surface Model (DSM). It contains not only relief in the form of a digital relief model (DRM), or digital terrain model (DTM), but also the surface of vegetation, buildings, and other man-made infrastructure.

The relief surface in the form of 2D DRM is usually separated from other 3D objects. The details of the object mapping system are expressed by the level of detail (LOD). LOD-0 is a regional model containing only a digital terrain model, LOD-1 contains a city/block model without roof superstructures, LOD-2 includes information about the shape of the roof and texture, LOD-3 contains detail information about the geometry (architecture) of objects and LOD-4 is an interior model which can describe the interior space of buildings and objects. In addition to above-ground elements, the subsurface structures are part of the 3D model as well. The subsurface objects are represented by geometric elements (points, lines, planes) in vector format with three spatial coordinates.

The methodology of processing of the 3D model of underground mines (Figures 2–7) in the Gelnica territory was as follows [34]:

- The floor plan had been georeferenced and then placed upon the existing maps - geographic coordinates were assigned to the old mining works.
- The individual mining horizons (adits) were digitized using the ArcGIS software application; in order to be able to work with them further, each horizon had been assigned one layer.
- Each horizontal layer had been assigned a specific altitude by analyzing the elevation profile layers, which in the next step lead to conversion from a 2D model into a 3D model.
- The next step represented modeling of vertical segments—adits, which had been defined as linear objects with two altitude values according to their endpoints.
- The connecting corridors and similar linear diagonally inclined underground structures had been defined in a similar way, where the endpoints were assigned not only different altitudes but various positions as well.
- The final step consisted of coloring of individual parts of the mine works according to their depth and redefining their size to obtain a representative model and to maintain its informative value as high as possible.

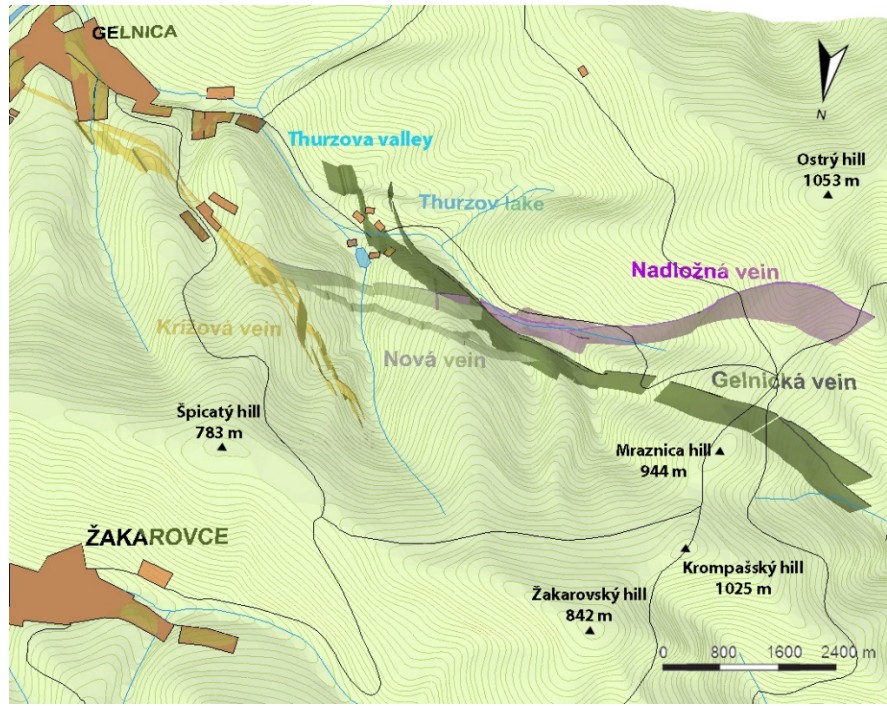

**Figure 2.** Overall 3D view displaying the Thurzova Valley vein system (situated north-west of Gelnica town) with a number of galleries and shafts excavated in the early 18th century.

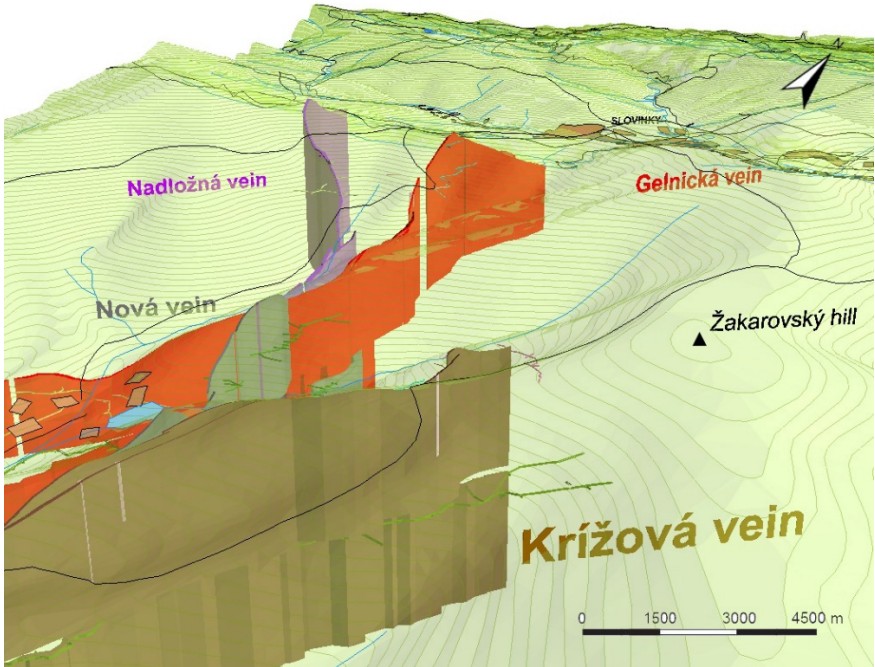

**Figure 3.** 3D view of the northern part of the vein system at the end of the Thurzova Valley with the mining underground space (2nd half of the 18th century).

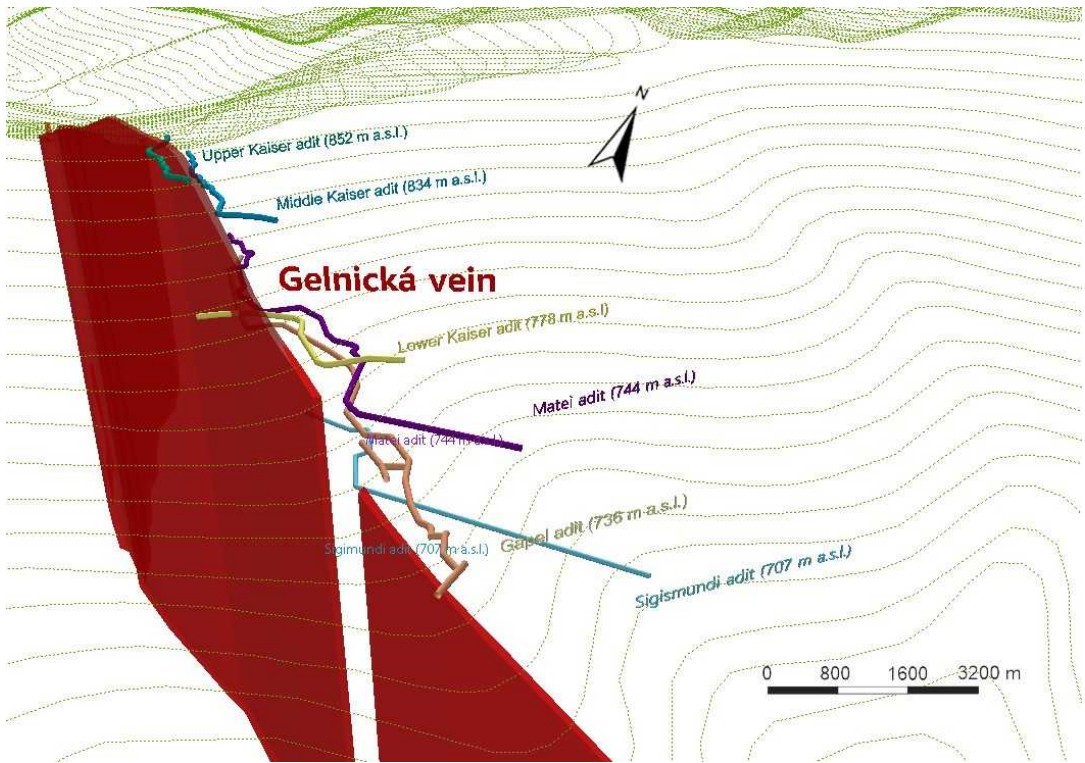

**Figure 4.** 3D view of the western part of Gelnická vein at the end of Thurzova Valley with inaccessible mining underground (condition in the second half of the 18th century).

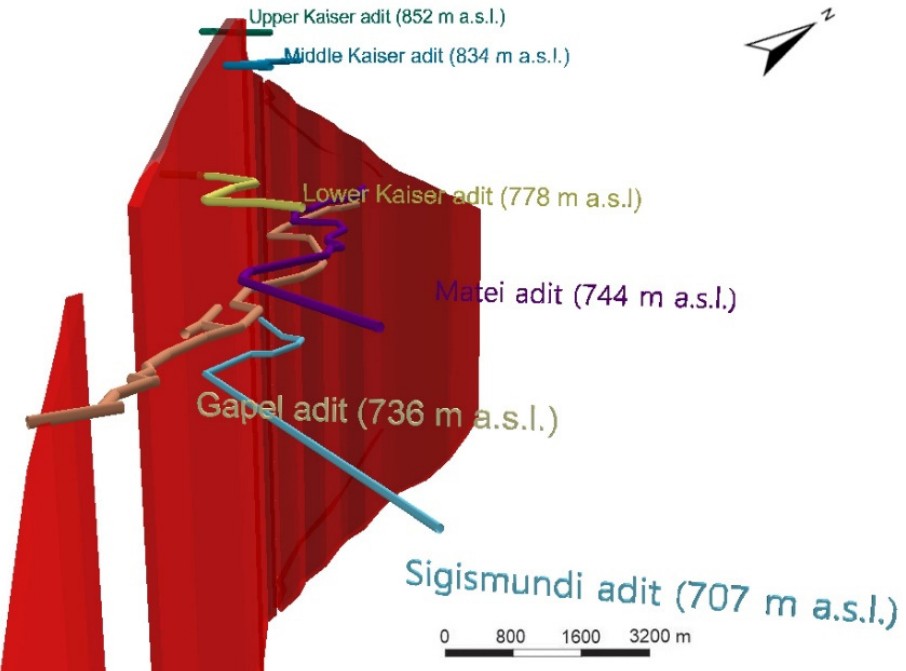

**Figure 5.** Detailed 3D model rotated 30 degrees east.

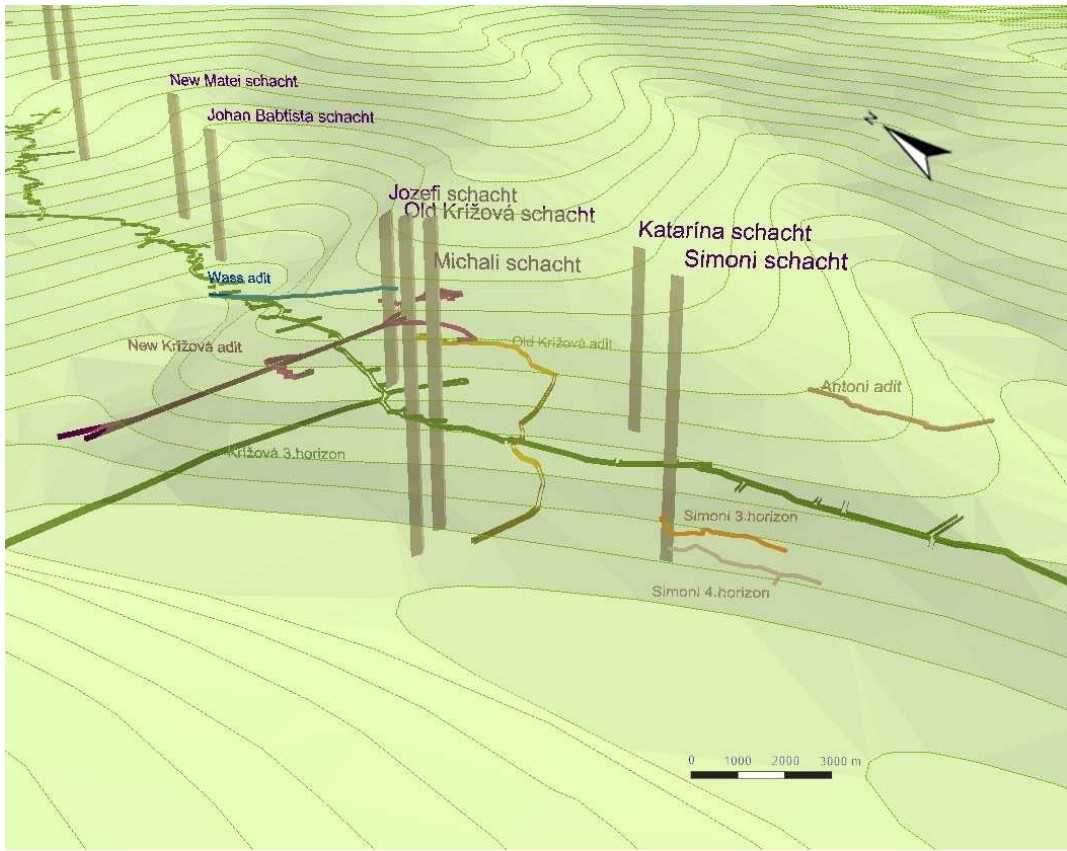

**Figure 6.** 3D view of corridor system and mine shafts in individual horizons on Gelnická vein in the central part of the Thurzova Valley (second half of the 18th century).

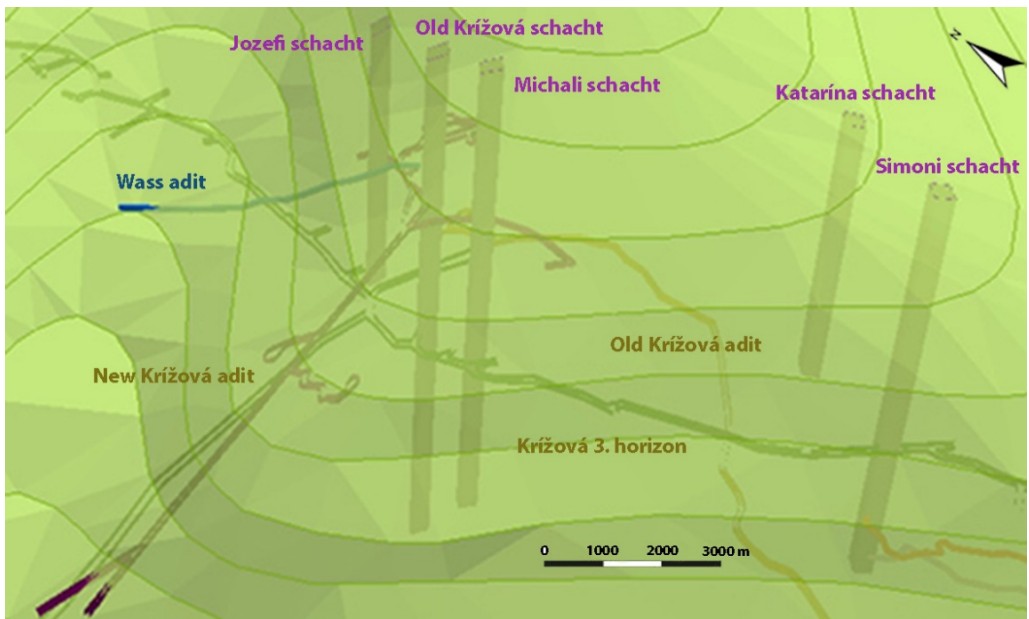

**Figure 7.** Localization of mining spaces below the earth's surface and thanks to the main image with 3D visualization are these mining spaces visible and 'accessible' to tourists after the overburden has been made transparent.

Many other modern technologies are currently being used to map underground mining areas and then create 3D models, such as mine surveying, photogrammetry, laser scanning, automatic laser scanning, terrestrial laser scanning [41–48].

## 4. Results

The computer modeling of the historical underground mining objects representing part of the contemporary landscape was implemented using the example of mines in the hinterland of the former free royal town of Gelnica. Presentation of the underground landscape elements had been created through natural landscape elements—ore veins and anthropogenic landscape elements—the mining underground.

The ArcGIS software package is an appropriate tool to create a geological model of the bearing and for modeling of underground ore veins, using which it is possible to create a detailed model for the needs of mining tourism with sufficient information and data mainly from exploration wells and archival documents [33]. Before the establishment of the 3D model, a complex 2D map basis over the ore veins had to be handled at first.

The formation of ore veins maps in two-dimensional space could be made only with successfully georeferenced maps [31,32,49–51]. After georeferencing, the individual maps from different provenances (mainly archival manuscripts of ore veins maps) had been mutually translated (ensuring an overlap of identical parts), allowing us to obtain a complete digital map of four ore veins around Gelnica.

Since the model was not created for professional use in geological surveys and given the diversity of the underlying archive sources, we had to adopt a small number of simplifications in the 3D model of the four ore veins. Ore veins had to be simplified to plate bodies in terms of their spatial characteristics. Digging depth of each vein is only estimated and may vary in reality in 250 m range (depending on the data from the last exploratory wells). Ore veins enter the surface in places where pings, dumping fields, and portals of old adits are found. Each vein is characterized by its inclination. All Gelnica veins are characterized by steep or very steep inclination in the range between 60–90°. Since the ArcGIS solution for modeling of the real inclination of veins is very difficult, we have decided to consider all veins as vertical, with an inclination of 90°. The actual course of the veins is documented in two horizontal layers at 340 and 560 m above sea level [18].

After geological veins modeling, the majority of which had been exploited in the past, the 3D model of the underground space attached to these veins had been created. Historical mining maps showing a floor plan of mining works were used in addition to the elevation profile maps of the underground. Mining maps from the 18th and the 19th centuries had been used as the basic layers, which were processed into a single digital map (2D model) of the underground mining spaces in the Gelnica territory using the identical methodology as during the processing of geological veins.

Currently, the underground of Gelnica can be seen on a static computer 3D model created as an output from ArcGIS Scene, which forms part of ArcGIS. It is a VRML file, a graphic format designed primarily to describe three-dimensional scenes, containing both active and passive objects used, for example, in virtual reality applications. This format allows you to look into the virtual underground of a surveyed area using information technology.

On the basis of several years of systematic research in the field of mining tourism and virtual mining, which were published in several scientific papers (the works are listed in the literature), we compiled a synoptic table summarizing strengths and weaknesses of virtual mining tourism as a new progressive direction within mining tourism. We focused on the assessment of the socio-cultural dimension, education, promotion, economic benefits, environmental impacts, modern technologies, and the political and legal environments (Table 1).

**Table 1.** Analysis of strengths and weaknesses of virtual mining.

| Virtual Mining | Strengths | Weaknesses |
|---|---|---|
| **Socio-cultural dimension** | - Access to experience also by physically disabled people.<br>- Improving the name of mining in the general public.<br>- Building partnerships between industry and the local community.<br>- Increasing local patriotism.<br>- There is a need to preserve the industrial heritage. | - It does not directly create new job positions.<br>- It does not directly create a secondary tourism offer.<br>- Impact on the social environment of the resident population (autochthonous population). |
| **Education** | - New attractive forms of education.<br>- Interest growth in mining and the history of mining in the young generation.<br>- Development of cultural values.<br>- Raising the level of education, building awareness and a positive relationship to the heritage. | - The need to have key competences in IT.<br>- The need to own hardware. |
| **Promotion** | - Strong promotional experience—creating motivation for a real visitor.<br>- Easily accessible advertising—web, apps.<br>- Support for existing mining tourism. | - It is a promotion not for the product itself but for the area that the virtual mine presents.<br>- It represents higher investments and expensive operating technologies. |
| **Economic benefits** | - Secondary bring visitors to the area.<br>- Generating traffic revenue.<br>- Compensation for declining industrial production.<br>- Diversification of economic sectors.<br>- Onset of the economic boom of the region or area.<br>- Increasing employment.<br>- Impact on the balance of payments of the region. | - They do not present direct support of the area.<br>- They can substitute the reconstruction of real monuments.<br>- They present higher investments and expensive operational technologies. |
| **Ecological impacts** | - Threats created in the past can be identified.<br>- They improve environmental awareness and relationship to nature.<br>- They present environmental changes (historical evolution), which cannot be seen in the real environment. | - There are no positive changes in the environment—conservation. |
| **Technology** | - Interconnection of historical facts and modern technologies.<br>- Possibility to use almost all forms of modern technology. | - Rapidly changing technology a constant need for change.<br>- In some cases, a network connection is needed. |
| **Political and legal environments** | - Government financial assistance; the number of subsidies provided by local authorities;<br>- safety, health care, and protection; building of transport infrastructure; support of local products, brands, and inhabitants. | - Settling of mining brownfield areas by marginalized population groups without property settlements. |

## 5. Discussion

The spatial 3D models of the historic mining landscape in the surrounding of the town of Gelnica brings examples of new possibilities for the presentation of mining sites in the mining tourism.

These models enable mining tourism clients to make a virtual visit and to see not only the existing, but the inaccessible mining objects (e.g., mining underground), and the vanished mining objects or elements contained 'in the real historical' landscape through the Internet technologies (of course, processing of information via the Internet was not the aim of the presented paper). These possibilities enable the development of mining tourism in small mining sites that have only a limited range of attractive constructions in situ. Computer technology allows the client to visit an underground mine, even where it is not possible in terms of safety, or to view underground spaces that have already collapsed.

Digital 3D models enable mining tourism clients to inspect mining works from each side or angle, and not only move in the underground areas, but to move through the entire mine from the surface to the deepest point. The client can make this journey not only from the comfort of his own home but directly in the landscape as well. Models allow viewing the landscape (the underground) from the current point of view. Another advantage of the technique comes from the fact that it is possible to look at the landscape, the underground, or at a different object in different timeframes if it is allowed by the technical and content processing of the models.

Next, the advantage of 3D computer modeling is that, in addition to the primary visual modeling obtained and in the composition of directly visible and measurable parameters of the mining landscape and its objects, it is possible to derive secondary parameters invisible in the historical materials. However, this process (research) requires expertise; therefore, 3D models are also used in the scientific sphere, particularly in monetaristic, mining geomorphology, history, environmental history, etc.

When we look at our 3D model in the context of other available solutions, we can summarize its advantages and disadvantages. Compared to other models, our model achieves high accuracy based on historical data and better visualization of objects in the final model. However, we must admit that historical maps scanned by a professional flatbed scanner achieve higher accuracy and lower noise. Higher accuracy can also be seen in the final model. In terms of usability, our solution outperforms other methods in obtaining real GPS coordinates of the places where there were significant objects in the past, which we only learn about from historical maps. Our method achieves an accuracy of approximately 1 m, which is sufficient for tourism. The integration of other objects, including the environment, into the output model, is to be part of follow-up research. Despite the above, our solution provides scope for new research opportunities in this field at a minimal cost.

The resulting form of final computer 3D models of the Gelnica historical underground is also available in scientific deliverables [34,35]. The second phase and vision of the whole project were to make the online virtual geological and mining underground available for visitors of Gelnica on site. What it actually means is that the client will walk the surface of the area. The tourist will have a mobile device (tablet, mobile phone, laptop, netbook, etc.) in their hand. The device, however, must have an integrated navigation system (in the initial stage GPS is being considered) and an active connection to a mobile data network (WiFi Hot spot is only considered the last resort). The exact location of the tourist would be known through the GPS in the device. Subsequently, the position would be recalculated by software to make it clear on the model where on the surface the tourist is. The mobile device would download a 3D computer model (in other words, it's relevant part, depending on the location of the device) with mobile data, which the tourist could view (rotate, zoom in) in real-time. The mobile device would show the underground space located directly under the tourist at the very moment. The model could be supplemented, modified and updated as required or according to new archival and present-day materials. This opens a huge space for presenting historical as well as contemporary objects in individual thematic layers with the possibility to present different time periods.

Similarly targeted web portals are for example following platforms which have been used for several years: Montanistics-Multimedia guide to mining tourism (Available online: https://www.montanistika.eu/), Slovak mining route (Available online: http://slovenskabanskacesta.sk/mapovy-portal/), Barbora route (Available online: https://barborskacesta.com/), Map of old mining burdens in Slovakia, prepared by The Geological Institute of D. Štúr in Bratislava (Available online: http://apl.geology.sk/geofond/sbd/) and others. However, these platforms use a lower display resolution mode.

3D models of historic mining landscape rank among the digital heritage of humanity, which consists of unique resources of human knowledge or expressions. It includes cultural, educational, scientific, administrative or technical, medical, or other kinds of information created digitally or converted into digital form from the existing analog resources. Where resources are "born digital", there is no other format but the digital object. The purpose of creating and preserving the digital heritage is to preserve the past and current events, objects, or intangible information to the public. Digital heritage should be a key element of national preservation policy, archive legislation, and obligatory or voluntary deposits in libraries, archives, museums, and other public repositories. The digital heritage is inherently unlimited in scope of time, geography, culture, or format and is specific to a particular culture, while being potentially accessible to every person in the world. Digital heritage has become a subject of protection of the UNESCO [15].

Based on the analysis in Table 1, it can be concluded that the main strengths of virtual mining tourism lie in support of (regional) tourism development with year-round use. As virtual mining tourism also presents the tangible cultural heritage, it brings synergy between the cultural values and the economic interests of tourism actors in order to increase their competitiveness in the market. Utilizing the potential of mining heritage to create new digital products, this type of tourism opens up new employment opportunities, helping to balance regional disparities and foster the sustainable development of the territory. Another strength of virtual mining tourism is a comprehensive system of marketing communication and strategy. Virtual mining tourism has a strong educational character with high informative value, and thanks to the web platforms and applications, it is used nationwide while covering all aim groups participating in tourism. In comparison to strengths, weaknesses are not so distinctive. However, their identification is important in order to adopt appropriate measures to mitigate their impact or to eliminate them. The most significant weakness is the lack of capital of tourism entities and also the initial investment in hardware and software equipment, which can become outdated quickly. The second weakness of virtual mining tourism is reflected in social contexts. In this sense, there is a need to adopt pro-growth measures that will stimulate the activities of all stakeholders in tourism in order to create working opportunities for an autochthonous or marginalized population, which will have a positive impact on the overall development of a territory.

## 6. Conclusions

It is undisputed that the digitization of old maps brings a lot of relevant historical information, which is becoming wider and irreplaceable position not only in the scientific but in the social sphere as well. Computer modeling in a GIS environment, based on the relevant historical documents and maps, as well as on a systematic historical and field research, brings new attractive displays in the form of different, often animated 3D models.

In recent years, professional computer reconstructions increasingly find their place in tourism, including mining tourism. These visualizations in the form of images, 3D models, or even films or holograms virtually transport the clients to a real historical period; thus the digital computer modeling allows them to travel in time and space.

The use of 3D models and visualizations of mining (inaccessible and non-existing) underground spaces, in practice, is of a popular-educational character. The models and visualizations are used for the purposes of the general public (especially in tourism or mining tourism), locals (with an emphasis on the marginalized groups of people who often inhabit these localities, e.g., Gemer, Spiš regions). The use of three-dimensional modeling could be also purely educational, in the teaching process at elementary and secondary schools as motivation tool for pupils (excursions, trips), but also as teaching aids in the terrain while educating new experts in the field of mining, geology, geography, or history at universities. Last but not least, it is possible to make extensive use of the work results for the needs of state or local authorities and other institutions. Moreover, these models can be used as background materials for various papers, reports, projects, and expertise, whether for social, economic, landscape protection, or conservation purposes.

Such models have their main use in promoting new trends in tourism—mining tourism, which is currently an important source of sustainable development of tourism in Europe.

**Author Contributions:** Conceptualization, P.H. and B.G.; methodology, P.H., and M.M.; formal analysis, L.H.; investigation, P.H., B.G. and M.M.; resources, P.H. and D.T.; data curation, D.T. and L.H.; writing—original draft preparation, P.H. and D.T.; writing—review and editing, D.T.; visualization, L.H. and M.M.; supervision, B.G.; project administration, P.H.; funding acquisition, B.G. and P.H. All authors have read and agreed to the published version of the manuscript.

**Funding:** This research was funded by grant number APVV-18-0185: Land-use changes of Slovak cultural landscape and prediction of its further development, and project VEGA 1/0236/18: Environmental aspects of mining localities settings in Slovakia in the Middle Ages and the beginning of Modern history.

**Conflicts of Interest:** The authors declare no conflict of interest.

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
