# Peer review of "Modeling of Vanished Historic Mining Landscape Features as a Part of Digital Cultural Heritage and Possibilities of Its Use in Mining Tourism (Case Study: Gelnica Town, Slovakia)"

_resources, doi:10.3390/resources9040043_

Round 1

Reviewer 1 Report

No further request

Author Response

All comments and suggestions recommended by the Reviewer 1 in the 1. round of the revision were accepted. The Reviewer 1 has no futher requests.

Reviewer 2 Report

Dear Authors,

I am pleased to see that all major reviewers suggestions have been incorporated. A lot of effort has been made to improve the manuscript.

The current version of the manuscript has certainly been substantially improved as regards both the text and the figures.

I have just two more suggestions:

-in my opinion the section 2 Aim of research can be eliminated and all the text of lines 67-81 can be moved in the section Introduction

-In the Figure 4a and 5 some writings appear too small and difficult to read; please improve their readability

Author Response

This manuscript is a resubmission of an earlier submission. The following is a list of the peer review reports and author responses from that submission.

Round 1

Reviewer 1 Report

Interesting manuscript and practical application of modern technologies for the purposes of mining tourism development.

Bellow, please find the reviewer‘s comments:

1) Introduction:

- the purpose of the article/research should be described better. Is it just visualising the mining heritage or also the assessment of the suitability of this activity in relation to tourism development?

- lack of references when describing the historical aspects

- lack of reason why the 3D visualisation is important for tourism development (I know that it is „obvious“, but I suppose that for a reader it would be interesting to add information e.g. about the legislative aspects – I suppose that it is complicated to get access especially to the underground mines - due to the security or other reasons. This would also support the necessity of the virtual mapping or 3D visualisation of mining heritage

- two last paragraphs would be better to assign to the „Material and methods“ chapter or it would be possible to create a specific chapter „Study area“ where the Gelnica surroundings and its mining aspect would be described more in detail (including its mining history)

2) Methods

- this chapter needs to be reorganised. At first, there should be a conceptual background for mining heritage and mining tourism (because mining heritage is a resource for mining tourism). This is mentioned later in the text (lines 110 and further), but generally, there is a lack of relevant resources for this conceptual background. Please, add more relevant references.

- in my opinion, the first and the second paragraph fits better to the Introduction

- line 92 and further – please re-structure the paragraph, it is a very long sentence

- are there any other methods of mapping underground mining spaces? For an overall view, it would be good to mention them.

- the technical description is sometimes lengthy, but acceptable.

3) Results

- the Figures should be more illustrative – it is not clear what is displayed there. Also, the scale, legend and eventually topographical map of the surroundings should be presented.

- I think that Table 1 is very important and it maybe should be stressed and discussed more in the Discussion chapter.

4) Discussion

- this chapter is very poor as it does not confront the results with already existing visualisations or analysis of the suitability of 3D visualisation.

- there are no references, no comparisons with already mapped sites. I suppose that your case is not the only case of mapping or visualising mining heritage, maybe in the surrounding countries, you can find other examples.

- please add relevant resources and literature or reconsider this chapter

5) Conclusions

- It would be good to add practical aspects and directions of further use of the method: did you present your work to local stakeholders? Is the visualisation available on the web pages of municipalities or regions both for locals and for tourists? Or did you somehow use the results of your work in local schools? I think this is one of the main purposes of the research – practical applications for tourism and eventually education. You could at least mention these future directions and possible use of the products of your research.

To conclude, the article is not bad, but it needs some arrangements, additions and corrections, especially in the conceptual background, graphical outputs, discussion and future directions or practical use of the results. After incorporating this, I recommend the article to be published.

Reviewer 2 Report

The topic of the paper is very interesting and undoubtedly of huge impact for readers. However it's not clear the main goal of the research that (as said by the authors) should describe "a methodology for 3D model processing". A GIS application is described but no other digital applications. A wide part of the text tells how should be necessary to apply, e.g., the virtual reality but nothing about the methodological approach for the realization of the relating 3D models in relation to the characteristics of the object (mining landscapes). Nothing about the integration and interaction of digital data using different processing platforms. The GIS application example is too simple and doesn't allow to understand the whole structure of the project. Quite interesting is the table 1 about project  strengths and weaknesses. But it's not enough on a scientific point of view. In conclusion, there is a part (about the site features) that is good and another one (about 3D modelling) that is too generic (without a real scientific experimentation) and that provides only intentions .

I suggest to change the weights of these two parts pointing the attention more on the first one.

Reviewer 3 Report

Dear Authors,

the manuscript concerning the methodology for 3D model processing of historic mining landscape  to use in mining tourism  is certainly of interest, especially because it answers the question of how to make fruible to the general public those sites of cultural heritage  which are inaccessible or accessible, for safety reasons, only to experts with specific equipment (mining, caves...).

However, I think that the structure of the manuscript must be improved and that in some parts it is incomplete as regards both the text and the figures.

My main suggestions are listed below.

Introduction

The introduction needs to be expanded and improved. In my opinion it should contain an overview, with relevant references (currently they are completely missing), of the existing work on 3D computer modeling used in mining  tourism and similar contexts.

Materials and Methods

Some parts contained in this section could be moved in the introduction (see for example lines 59-74).

In lines 191-193 it is stated: By the creation of 3D models for the needs of geotourism and mining tourism we followed the methodology developed by M. Molokáč and L. Hvizdák in collaboration with prof. P. Rybár at the  Faculty BERG of the Technical University in Košice [24-30]. It is not explained, however, what specifically the authors of this manuscript did (How were these 3D models created? Using which supports, materials?). In my opinion, the methodology used in the present work should be specified point by point.

Results

Lines 223-227: I think this part is not a result, it should be moved to the Materials and Methods section, or alternatively you should consider entering a specific section/sub-section “Study area”.

Lines 254-269: I think this part is not a result, it explains the methodology used and should be moved to the Materials and Methods section.

In this sectionmore emphasis should be given to the results obtained in terms of quality of the created 3D models and of their potential use. To make clearer the results obtained and make them better appreciate to the reader, more figures should be shown, offering views from different angles and with different enlargement/zoom.

Discussion

In my opinion this is the best part of manuscript. It is well written, but not sufficiently supported by the results as they are currently presented.

Figures

Scale and orientation (north) are missing in all the figures. Please add them.

Figures 3 and 4: the legend is missing. Please add this.

To make the results obtained better appreciable, the number of  figures should be increased.